# Therapeutic Potential of α-Crystallins in Retinal Neurodegenerative Diseases

**DOI:** 10.3390/antiox10071001

**Published:** 2021-06-23

**Authors:** Ashutosh S. Phadte, Zachary B. Sluzala, Patrice E. Fort

**Affiliations:** 1Department of Ophthalmology and Visual Sciences, University of Michigan, Ann Arbor, MI 48105, USA; Ashutosh.phadte@jefferson.edu (A.S.P.); zsluzala@umich.edu (Z.B.S.); 2Department of Molecular and Integrative Physiology, University of Michigan, Ann Arbor, MI 48105, USA

**Keywords:** alpha-crystallin, neurodegeneration, recombinant proteins, oxidative stress

## Abstract

The chaperone and anti-apoptotic activity of α-crystallins (αA- and αB-) and their derivatives has received increasing attention due to their tremendous potential in preventing cell death. While originally known and described for their role in the lens, the upregulation of these proteins in cells and animal models of neurodegenerative diseases highlighted their involvement in adaptive protective responses to neurodegeneration associated stress. However, several studies also suggest that chronic neurodegenerative conditions are associated with progressive loss of function of these proteins. Thus, while external supplementation of α-crystallin shows promise, their potential as a protein-based therapeutic for the treatment of chronic neurodegenerative diseases remains ambiguous. The current review aims at assessing the current literature supporting the anti-apoptotic potential of αA- and αB-crystallins and its potential involvement in retinal neurodegenerative diseases. The review further extends into potentially modulating the chaperone and the anti-apoptotic function of α-crystallins and the use of such functionally enhanced proteins for promoting neuronal viability in retinal neurodegenerative disease.

## 1. Introduction

The retina is a metabolically active tissue which serves to receive and translate incident light into neural impulses for their relay to the visual cortex of the brain. Anatomically, the retina is a highly complex tissue with diverse neuronal subtypes within its circuitry serving a distinct set of functions. Different cell types of the neural retina undergo evolutionarily driven adaptations to better serve their function while fulfilling the characteristic optical requirements of the tissue. For terminally differentiated cells such as the retinal neurons, their axons are inherently predisposed to high exposure to free radicals, mechanical compression, as well as photo-oxidative damage [1]. The adaptations generate energy mismatches that are associated with a high energy requirement and therefore influence their high metabolic rate.

Retinal photoreceptors (PRs) and ganglion cells (RGCs) are the primary cells affected in retinal neurodegenerative diseases such as age-related macular degeneration, retinal dystrophies, glaucoma, and diabetic retinopathy. This observed susceptibility is in part due to their unique characteristics and the nature of the surrounding environment in which they have to perform their function. Over time, PRs and RGCs face an exponential increase in the incidence of stressors, including increased oxidative stress, altered energy homeostasis, accumulation, and aggregation of misfolded proteins and detrimental lesions in cellular DNA [1,2]. Under homeostasis, multiple modules of the cellular quality control machinery such as the DNA mismatch repair (MMR), base excision repair (BER), ubiquitin-proteasome pathway, and lysosome mediated autophagy function to prevent adverse effects of accumulation of DNA lesions, proteinaceous aggregates, and toxic byproducts. Aging results in the progressive decline in the efficacy of this quality control machinery, eventually compromising neuronal viability. The observed progressive neurodegeneration is a common feature and is considered as a hallmark associated with diseases affecting the CNS such as Alzheimer’s disease (AD, [3,4]), Parkinson’s disease (PD, [4]), Huntington’s disease (HD, [5,6]), Amyotrophic lateral sclerosis (ALS, [7]), and Crutzfield–Jacobs disease (CJD, [8]). Injury and pathophysiologies affecting the neural retina—retinopathies and glaucoma—also manifest clinical features of neurodegeneration as observed with diseases of the CNS, with a progressive loss in the number of neuronal cells, compromising visual acuity and quality of life. Targeted therapies aimed at delaying the neurodegenerative aspects of these diseases have therefore been a much sought-after area of scientific research.

Cellular exposure to a multitude of environmental (light, radiation, heat shock), physiological (growth factors, inflammatory signals, and hormones), and pathological (PAMP, ischemia) stimuli have often been shown to result in an increase in expression of chaperone proteins, highlighting their key role in maintaining cellular homeostasis. This broad class of proteins can be distinctly divided into two classes. The first class includes the larger ATP-dependent heat shock proteins (HSPs) that function as ‘foldases’ to refold misfolded proteins in an ATP dependent fashion. Additionally, HSPs also help in targeted degradation of misfolded proteins by the ubiquitin-proteasome system and autophagy. Prominent examples of this class include HSP100, HSP90, HSP70, HSP60, and HSP40. The second class represents small heat shock proteins (sHSPs), which are primarily ATP-independent ‘holdases’ that bind and stabilize unfolding protein intermediates and facilitate their refolding by HSPs. Ten members of the sHSP family have been identified in humans—HSPB1-10. Members of the α-Crystallin family of sHSPs—αA (HSPB4) and αB (HSPB5)—have been extensively characterized in the eye lens as molecular chaperones. Over time, the lessons learned relative to their chaperone function in the lens has been key in highlighting their potential neuroprotective role in the context of neurodegenerative diseases of the central nervous system. In the last decade, multiple studies have shown the involvement of α-Crystallins in retinal homeostasis and disease (summarized in reviews [9,10,11,12,13]). The current review evaluates the therapeutic potential of α-crystallins and its derivatives, especially αA-crystallin, in the treatment of neurodegeneration associated with retinal pathologies.

## 2. The Chaperone and Cytoprotective Function of α-Crystallins

Lens crystallins were first described by Mörner in the later years of the 19th century as constituent proteins that make up the crystalline lens. The three classes of crystallins—α, β, and γ—account for approximately ninety percent of the total soluble protein in the eye lens [14]. The α-crystallins, comprise of αA and αB subunits, are encoded by genes CRYAA (αA- on chromosome 21) and CRYAB (αB- on chromosome 11). Both isoforms share about fifty seven percent identity in their protein sequences [15,16]. It has been speculated that the two proteins have evolved as a result of ‘gene-sharing’, such that they serve both a structural as well as functional role in the eye lens [17]. 

Initially thought to be restricted to the eye lens, studies have corroborated a non-lenticular expression pattern for α-crystallins. CRYAB has been classified as a class I HspB gene due to its ubiquitous pattern of expression in a wide range of tissues such as the retina, heart, kidney, skeletal muscles, brain, and skin, with an observed increase in protein expression in response to stress [18,19,20]. CRYAA on the other hand was classified as a class II HspB gene, primarily thought to have been expressed in the lens alone, and that protein expression was not influenced by induction of stressors. Since then, multiple studies have reported αA-crystallin in several tissues, but especially so in the central nervous system (CNS) with many reports in the brain and the retina [21,22,23,24,25]. 

Investigation into the function of α-crystallins led to the discovery of their chaperone activity [26]. Both αA- and αB-crystallins were shown to function as molecular chaperones, efficiently preventing non-specific protein aggregation of ‘substrate proteins’ in vitro. The chaperone activity of α-crystallins was attributed as a principal mechanism behind the underlying maintenance of the transparency and the refractile properties of the eye lens [27,28,29]. Expression of α-crystallins in cells was shown to positively influence their viability under stress, suggesting that the observed chaperone activity of the proteins in vitro also translated into their cytoprotective ability under stress [30,31,32,33]. αA- and αB-crystallin expression has since been shown to correlate with increased cellular survival in the presence of external stressors such as UVA/UVB [34], staurosporine [35], etoposide [35,36], hydrogen peroxide [33,37,38], TNFα [37,38,39,40], and serum starvation/nutrient deprivation [22]. 

## 3. Mechanistic Analyses of the Anti-Apoptotic Function of α-Crystallin

A study by Pasupuleti et al. was instrumental in showing a direct dependence between the chaperone activity of αA-crystallin and its observed anti-apoptotic function [36]. Mechanistic analyses of the observed cytoprotective abilities of αA- and αB-crystallins suggested the involvement of multiple cellular targets through which the proteins mitigate their response. Human lens epithelial cells stably expressing αA- and αB-crystallins were more resistant to staurosporine-induced apoptosis via sequestration of pro-apoptotic molecules Bax and Bcl-Xs, preventing their translocation to the mitochondria [35]. The study also emphasized an appreciable increase in cellular viability of α-crystallin transfected human lens epithelial (HLE), retinal pigment epithelial (ARPE-19), and rat embryonic myocardial cells (H9c2) on exposure to etoposide and sorbitol induced hypertonic stress. Stable expression of αA-crystallin in Chinese hamster ovary (CHO) cells prevented Bax and Bim-induced apoptosis in comparison with vehicle controls. αA-crystallin expression also prevented doxorubicin induced apoptosis of CHO cells by preventing the activation of procaspase 3. In another study, expression of αA-crystallin was also shown to promote HeLa cell survival by increasing the activation of prosurvival protein kinase B (Akt) [36]. Studies by Mehlen et al. emphasized the efficacy of αB-crystallin in preventing TNFα-induced apoptosis in L929 fibrosarcoma cells, a TNFα sensitive cell line [33]. Expression of αB-crystallin also conferred resistance to L929 cells against exposure to menadione and hydrogen peroxide mediated oxidative stress. Further insights into the mechanistic details revealed an increase in cellular glutathione levels as a consequence of sHSP expression, which thereby mediated the observed resistance to oxidative stress [40]. Expression of αA- and αB-crystallin was also shown to relate with an increase in cellular glutathione levels in RPE cells under hydrogen peroxide induced oxidative stress, indicative of a conserved mechanism associated with the observed retention in cellular viability [41]. Stable expression of αA- and αB-crystallins in human lens epithelial cells was shown to prevent UVA irradiation-induced cell death. While expression of αA- and αB-crystallins promoted HLE survival, functional analyses of the observed cytoprotective effect identified two distinct mechanisms of action. αB-crystallin mediated cell survival was found to have been mediated through the attenuation of RAF/MEK/ERK cascade, whereas αA-crystallin expression prevented cell death through an upregulation of the PI3K/Akt pathway. In addition, UVA irradiation of HLE cells stably expressing αA- and αB-crystallins also resulted in an upregulation of PKCα, although mechanistic details of this pathway in HLE survival remain to be investigated. Studies from our lab have also highlighted the neuroprotective efficacy of αA-crystallin in diabetes-induced neurodegenerative stress in the retina. αA-crystallin expression resulted in the attenuation of metabolic stress-induced activation of ER stress specific markers eIF2α, CHOP, BIP, and PDI in retinal neurons [22]. Studies also support an anti-inflammatory potential of αB-crystallin, where lentiviral expression of the protein attenuated NF-kB mediated expression of pro-inflammatory cytokines in THP-1 cells exposed to LPS [42]. A recent study from our lab is highly supportive of the anti-inflammatory potential of αA-crystallin. Expression of αA-crystallin in primary Müller glial cells effectively attenuates reactive gliosis under nutrient deprivation and diabetic stress, primarily through the mitigation of the NF-kB and inflammasome regulatory pathways [43]. Collectively, it can be surmised that even though the proteins share considerable similarity in their structure and function in vitro, their anti-apoptotic function can be exerted through a combination of specific and partially overlapping pathways and cell survival mechanisms. Figure 1 summarizes the different cellular targets through which both αA- and αB-crystallins mediate their anti-apoptotic function.

## 4. Post-Translational Modification of α-Crystallins and Modulation of the Chaperone Activity

The effect of post translational modifications (PTMs) on crystallin proteins has been extensively documented in the eye lens. Due to lack of active protein turnover mechanisms, lens proteins are subject to progressive accumulation of PTMs over time, resulting in loss of function, development of opacities, and ultimately, cataracts. Studies on human lenses showed an age-dependent increase in the accumulation of PTMs on α, β, and γ crystallins with time [12,44]. The specific effect of PTMs on the activity of α-Crystallins, however, is still not fully understood. PTMs such as deamidation, oxidation, glycation, AGE modification, phosphorylation, acetylation, and proteolytic cleavage have been shown to affect the chaperone activity of both αA- and αB-crystallins. Mass spectrometric analyses on diabetic lenses from human donors have shown an increased amount of truncated αA-crystallin with age [45]. In vitro studies on truncated αA-crystallins have shown that truncation influences the oligomerization and solubility of crystallin homo- and heterooligomers by increasing their oligomeric molar mass and influencing their rate of subunit exchange [46]. Cleavage of the terminal serine-serine peptide bond generated αA_1-172_ crystallin, and in vitro analyses of recombinant αA_1-172_ crystallin revealed its enhanced chaperone activity [47]. Interestingly, MS analyses of donor lens tissue revealed a two-fold increase in the levels of αA_1-172_ in diabetic donor lenses over non-diabetic controls [45]. Additionally, streptozotocin induced diabetes in rats was also shown to result in an appreciable increase in the levels of truncated αA-crystallin over experimental controls, suggesting that diabetic stress resulted in the upregulation of enzymatic processes involved in α-crystallin truncation. Up to sixteen amino acids from the C-terminal of αA- and αB-crystallin can be subject to truncation [12,45]. Since the C-terminal of the proteins has been shown to contribute to maintaining their solubility, truncation has also been thought to result in loss of solubility of chaperone-substrate protein complexes, thereby causing cataract.

Conversely, PTMs have been thought as an adaptive mechanism of the lens to promote the functional longevity of an already depleting chaperone ‘stock’ with age. Modification of α-crystallins by sugars and ascorbate have been shown to reduce its chaperone function via formation of pentosidine mediated crosslinking [48]. Modification of αA-crystallin by methylglyoxal (MGO), a dicarbonyl AGE byproduct present throughout the lens was shown to increase the chaperone activity of the protein [49,50,51]. Acetylation of α-crystallins has been documented in donor lens tissue ranging from 15–86 years [52]. Acetylation of αA-crystallin on K70 and K92 has been shown to increase its chaperone activity in vitro [53]. Acetylation of MGO-modified αA-crystallin was shown to further enhance its chaperone activity in vitro, as well as serve as a ‘suppressor’ mutation to alleviate the loss in chaperone activity of ascorbate-glycation modified αA-crystallin [52]. External supplementation of acetylated, MGO-modified αA- and αB-crystallins to CHO cells under thermal stress resulted in retention of cell viability and inhibition of caspase-3 activity. It could therefore be asserted that prior acetylation of α-crystallin may serve as a protective mechanism in the lens as a countermeasure to age-dependent PTMs that compromise the chaperone activity of the proteins. This hypothesis has been further supported by the demonstration that acetylated peptide derivatives of αA-crystallin better prevent experimental cataracts in rats when compared to unmodified controls.

Both αA- and αB-crystallins have been shown to be phosphorylated on serine and threonine residues [12]. Mass spectrometric analyses of water insoluble fraction of human lenses revealed phosphorylation of αA-crystallins on T13, S45, S122, T140, and T148 [54,55]. αB-crystallin has been empirically shown to be phosphorylated on S19, S21, S43, S45, S53, S59, and S76. Owing to its ubiquitous expression throughout the body, phosphorylation of αB-crystallin has been extensively investigated, and studies on αB-crystallin mutants mimicking phosphorylation at S19, S45, and S59, either individually or in tandem, have yielded complex results. Overall, ex vivo studies support the role of αB-crystallin phosphorylation in enhancing its anti-apoptotic activity. Cardiomyocytes expressing an alanine substituted ‘triple’ mutant of αB-crystallin (S19A/S45A/S59A) exhibited increased susceptibility to sorbitol and hypoxia induced apoptosis in comparison to cells expressing wild-type and phosphomimetic mutants [56]. Treatment of cultured rat astrocytes with a p38 protein kinase inhibitor SB203580 or the ERK1/2 inhibitor PD98059, two kinases responsible for phosphorylation of αB-crystallin, increased their susceptibility to ceramide and staurosporine induced apoptosis [57]. Substitution of S19, S45, and S59 to alanine prevented nuclear retention of αB-crystallin in a rat retinal pigment epithelial cell culture system following transfection [58]. Exposure of the transfected cells to MGO resulted in enhanced cell death, evidenced by immunoblotting analyses against cleaved caspases 2L, 3, and 7. While evidence for αA-crystallin have been scarce, a recent study from our lab recently demonstrated a key role of phosphorylation on the cytoprotective function of αA-crystallin in the retina. Our studies on retinal tissue samples from human donors with diabetes revealed a high level of basal phosphorylation of retinal αA-crystallin on T148, a modification that was dramatically reduced in diabetes [22]. Cell culture experiments showed that expression of the αA-crystallin phosphomimetic T148D resulted in an enhanced survival of R28 cells under metabolic stress. Furthermore, in vitro assessment of the chaperone activity of the αA-crystallin phosphomimetic T148D revealed a two-fold increase in its chaperone activity in comparison to the wild-type protein [in press]. These data suggest that phosphorylation of retinal αA-crystallin on T148 has a direct consequence on its neuroprotective function and support the increased therapeutic potential of the enhanced chaperone phosphomimetic protein.

## 5. Development of Peptide Mini-Chaperone Derivatives of α-Crystallin

The observation of the chaperone activity of αA- and αB-crystallins was succeeded by subsequent investigations into identification of the chaperone sites on the proteins responsible for their observed function in vitro. As with other members of the sHSP family, α-crystallins were hypothesized to interact with aggregating protein substrates via hydrophobic ‘patches’ distributed throughout their amino acid sequence. Proteomic analysis of recombinant αA- and αB-crystallin structure by 4,4′-Dianilino-1,1′-binaphthyl-5,5′-disulfonic acid (bis-ANS) highlighted ^50^QSLFR^54^ and ^79^HFSPEDLTVKVQDDFVEIHGK^99^ in αA-crystallin and ^75^FSVNLDVKHFSPEELKVKVLGDVIEVHGK^103^ as bis-ANS interaction sites [59]. Interaction of melittin with αA-crystallin resulted in the loss of its chaperone activity, as evidenced by the lack of suppression of ADH aggregation by αA-crystallin preincubated with melittin [60]. Subsequent proteomic analyses revealed the melittin binding sites corresponding to amino acid residues 13–21 and 71–88 in αA-crystallin and 70–88 in αB-crystallin as melittin binding sites. Synthesis and in vitro chaperone assay analyses of a peptide corresponding to αA-crystallin residues 70–88 and αB-crystallin residues 73–92 showed that the two peptide ‘mini-chaperones’ efficiently prevented non-specific aggregation of proteins in vitro. The study was the first of its kind to demonstrate the chaperone efficacy of a peptide derivative of α-crystallin [60], although still requiring demonstration of its therapeutic potential in vivo. 

To that end, studies assessed the efficacy of the peptide mini chaperones to promote cellular viability under stress, evaluating the association between the observed anti-aggregation properties and their anti-apoptotic efficacy. αA-crystallin-derived peptide chaperone, ‘mini-αA’ crystallin was shown to prevent the aggregation-induced toxicity of amyloid beta peptide (Aβ) in vitro. Administration of the peptide mini-chaperone greatly diminished Aβ-aggregation induced toxicity and cell death in rat pheochromocytoma (PC-12) cells, as validated by morphological and cell death analyses [61]. Like other peptide-based therapeutics, the relative efficacy of the mini-chaperone peptides over time has been challenged by issues pertaining to their availability in vivo on administration, namely resistance to serum proteases, half-life as well as target specificity. Several studies have attempted to address these shortcomings and have made considerable progress. As mentioned previously, acetylation of mini-αA crystallin on K70 and mini-αB-crystallin on K92 was shown to greatly influence their anti-apoptotic function in primary organ cultures from rat lens explants on exposure to calcimycin [62]. Additionally, intraperitoneal administration of both acetylated variants of the mini-chaperone peptides was shown to prevent the progression of cataracts in mice following exposure to selenite [62]. The study demonstrated for the first time, the efficacy of α-crystallin derived chaperone peptides as a tool to prevent non-specific protein aggregation associated with cataract. Mechanistic analyses described a multi-level effect of these peptide chaperones in inhibition of oxidative stress-induced apoptosis, protein insolubility, and reduction of caspase activity in response to stress. Administration of mini-αA and mini-αB crystallins was also shown to prevent tert-Butyl hydroperoxide-induced cell death of human fetal retinal pigment epithelial (hfRPE) cells. Sreekumar et al. further investigated the mechanistic details of peptide uptake in hfRPE cells and reported the involvement of SOPT1 and two Na^+^ dependent oligopeptide transport systems in the uptake of mini-α crystallin peptides, using ^3^[H] labeled DADLE and deltorphin II uptake studies [63]. The authors further added that housing mini-αB-crystallin in polycaprolactone (PCL) nanoparticles helped enhance its efficacy to promote survival of hfRPE cells on exposure to H_2_O_2_-induced oxidative stress. A peptide variant of mini-αA crystallin designated CP1 (mini-αA crystallin + amino acids 164–173 of αA-crystallin), designed to enhance peptide solubility, was shown to prevent oxidative stress-induced cell death in ARPE-19 and COS-7 cells [64]. Intravitreal administration of mini-αA crystallin was also shown to promote RPE cell survival on exposure to NaIO_3_. Specifically, the enhanced survival of RPE cells was associated with the attenuation of the UPR response by mini-αA crystallin, which further resulted in decreased levels of the ER stress chaperones CHOP and XBP1 [65], a mechanism similar to that recently described with full length αA-crystallin in the diabetic retina [22]. In an attempt to increase their chaperone efficacy in cells, cell-penetrating sequences have been appended to full length αB-crystallin, which in turn resulted in an increase in their sequestration in lens cells [66]. Raju et al. appended a cell penetrating sequence derived from the Ku70 protein to the N-terminus of mini-αA crystallin. The resultant peptide, designated CPP1, was shown to efficiently sequester inside COS-7 cells following treatment and promoted ARPE-19 survival under conditions of oxidative stress [67]. Subsequent biochemical analyses revealed that the CPP1 peptide was also able to prevent Aβ-induced cytotoxicity in ARPE-19 cells, thereby promoting its anti-aggregation potential [67]. More recently, Stankowska et al. have demonstrated the therapeutic potential of mini-αB-crystallin in promoting survival of retinal ganglion cells under conditions of hypoxic stress, thereby extending their potential functional significance for the treatment of glaucoma and diabetic retinopathy [68]. While additional work is needed, this report represents the first evidence of the capacity of mini-αB crystallin to effectively cross the blood retinal barrier. Cy7 labeled mini-αB crystallin, designated as ‘peptain-1′ was reported to be found within the retina following intraperitoneal administration. Functionally, systemic administration of peptain-1 also correlated with an enhanced survival of RGCs exposed to hypoxic and ischemic stress. Morphologic assessment suggested that peptain-1 administration restored the anterograde axonal transport in RGCs and prevented damage to their soma on exposure to stress. Together, these studies represent a strong case for the therapeutic potential of α-crystallin-derived peptides; however, there remains limitations to their efficacy, bioavailability, stability, and overall protective effect when compared to full length functionally enhanced chaperone proteins.

## 6. Functional Repertoire of αA- and αB-Crystallins in Ocular Health and Disease

In the retina, αA- and αB-crystallins have been shown to exhibit a somewhat differential pattern of expression, especially under stress. While both proteins have been reported to be basally expressed at low levels in healthy retinal tissue, a significant induction of their expression in different cells of the retina has been observed under conditions of stress and disease. On exposure to stressors, αB-crystallin was shown to heavily localize within the retinal pigment epithelium (RPE) and photoreceptors, whereas αA-crystallin predominantly localized within the inner retina and more specifically the ganglion cell layer and Müller glia [22,69].

Ever since αA- and αB-crystallins were found to express outside of the eye lens, multiple studies have sought to understand their role in retinal homeostasis. Deretic et al. first reported the presence of soluble αA- and αB-crystallins in retinal homogenates from frog retinal lysates. The study emphasized the involvement of both proteins in the transport of newly synthesized rhodopsin from the Golgi network to the photoreceptor cell membrane [70]. Subsequent studies in mice reported that retinal protein levels of αA- and αB-crystallin were found to be increased approximately three-fold following intense light exposure in comparison with experimental controls. The study attributed this observed increase to an adaptive mechanism of the retinal tissue to prevent cell death in response to intense light mediated oxidative insult [71]. Microarray gene expression analyses of mice retinas subjected to retinal tear revealed a modest increase in the levels of αA- and αB-crystallin at different stages post retinal injury, suggesting a distinct role played by the two proteins as a part of the wound healing process [72]. 

The role of α-crystallin in context of ocular disease was investigated in animal models of disease as well as through analysis of human donor samples. Analysis of drusen samples from donor tissue with age-related macular degeneration (AMD) by LCMS revealed a five-fold increase in the levels of αB-crystallin in comparison with donor controls [73]. Further histological analyses revealed a colocalization of αB-crystallin with drusen along the Bruch’s membrane from AMD donor samples [74]. While the exact role and implication of this finding remains unclear, the study regarded αB-crystallin as an important biomarker in the pathogenesis of AMD. The anti-apoptotic activity of αA- and αB-crystallins was also investigated in models of local inflammatory disease. Both proteins were shown to play an important role in the regulation of pro-inflammatory cascades, as demonstrated by studies highlighting an increased susceptibility of α-crystallin knockout mice to *S. aureus* induced endophthalmitis [75] and uveitis [24]. Interphotoreceptor retinoid binding protein (IRBP) peptide-mediated autoimmune uveitis resulted in a ~30-fold increase in the levels of αA-crystallin mRNA along with immunofluorescence analyses localizing αA-crystallin within the photoreceptors. αA-crystallin was shown to interact and sequester cytochrome c, suggesting its involvement in quelling mitochondrial stress induced apoptosis [24]. Upregulation of αB-crystallin was shown to prevent retinal cell death during inflammatory clearance in a model of *S. aureus* induced destructive autoimmune uveitis [75]. Expression of αA- and αB-crystallins was shown to decrease in correspondence to elevated intraocular pressure (IOP)-induced RGC cell loss in a rat model of glaucoma. Interestingly, early stage RGC cell loss at two weeks post treatment (RGC loss ~eight percent) was accompanied by a reduction in the expression of αA- and αB-crystallin transcripts by fifty percent in comparison to experimental controls [76,77]. Five weeks post treatment, mRNA levels of α-crystallins were higher in the treated cohort in comparison to experimental controls, although the authors reported a twenty percent reduction in RGC. Similar observations were reported in independent studies in other models of experimentally induced glaucoma, via hypertonic saline injection in the episcleral vein [78] and in DBA/2J mice [79]. It can be speculated that the neuroprotective efficacy of retinal αA- and αB-crystallins could be compromised by mechanisms, which suppress their transcription under glaucomatous stress, although evidence of such processes remains elusive. 

α-crystallins also exhibit distinct neuroprotective functions within the retina. Expression of αA- and αB-crystallin decreased in the ganglion cell layer of mice retinas following optic nerve transection (ONT). Relative survival of axotomized RGCs increased following expression of α-crystallins in comparison to sham transfected controls [25]. αA- and αB-crystallin knockout mice exhibited an increase in cell death and retinal degeneration on exposure to cobalt chloride, suggesting their role in the inherent protective mechanisms of the retinal tissue under hypoxic stress [80]. Furthermore, multiple studies also support the efficacy of externally administered purified α-crystallins in promoting retinal ganglion cell survival and axonal regeneration following optic nerve crush, both in vitro and in vivo (discussed in detail in later section). Retinal samples from mice models of diabetes-streptozotocin induced diabetes [81], spontaneously diabetic Otsuka Long-Evans Tokushima Fatty (OLETF) rat model [82], and high fat-induced diabetes [83] revealed a marked increase in the levels of αA-crystallin within the retina in comparison to controls, whereas αB-crystallin expression was globally increased in the retina, heart, brain, and muscle [84]. Induction of diabetes using streptozotocin was shown to have altered the ability of αA- and αB-crystallins to interact with and sequester pro-apoptotic molecules, Bcl-Xs and Bax [81]. A more recent study from our lab emphasized a significant elevation of αA-crystallin in retinal cross sections from human donors with diabetes and diabetic retinopathy when compared to non-diabetic controls [22]. This study, reminiscent of the impact of aging on the function of αA-crystallin in the lens, reported that the observed increased expression of αA-crystallin correlated to an altered PTM profile of the protein modified by the induction of diabetes. Together, our observations strongly suggest that although an increase in the expression of α-crystallins in diabetes reflects their neuroprotective potential in preventing retinal cell death, disease progression is associated with an alteration of their inherent neuroprotective efficacy, eventually compromising protein function under chronic disease conditions. 

## 7. Systemic versus Local Administration of α-Crystallin

The overwhelming wealth of evidence supporting the anti-apoptotic and cytoprotective abilities of α-crystallins has warranted studies exploring their application as neuroprotective agents. Due to their inherent involvement in multiple cellular functions, studies have investigated the potential of both systemic and localized administration of purified bovine α-crystallin. One of the first studies to support the systemic administration of α-crystallins investigated their effect on systemic inflammation. Supplementation of α-crystallin to primary cells isolated from mice exposed to silver nitrate was able to suppress key elements associated with systemic inflammation [39]. Consistent with the observed effect ex vivo, intraperitoneal administration of α-crystallin purified from bovine lenses efficiently prevented silver nitrate-induced inflammation in mice [38]. Mechanistically, α-crystallin supplementation was able to prevent systemic inflammation in the CNS in mice exposed to silver nitrate, as evidenced by the relative levels of GFAP and NF-kB within the neocortex and the hippocampus [85]. Similarly, intravenous administration of αA-crystallin was shown to modulate the inflammatory response associated with experimental autoimmune uveitis (EAU) by suppression of levels of pro-inflammatory cytokines [23]. Although αB-crystallin was unable to influence the pro-inflammatory signal profile associated with EAU, recombinant αB-crystallin administration was shown to efficiently prevent tissue damage and improved motor skills in mice following spinal cord injury [86]. Intravitreal administration of αB-crystallin was also shown to alleviate photoreceptor degeneration in C57BL/6 mice on exposure N-methyl-N-Nitrosurea (MNU) [87]. The independent and specific effects and functions of the two α-crystallin proteins further highlight their respective therapeutic potential in different ocular diseases.

Like their minichaperone peptide derivatives, systemic administration of full-length α-crystallins has its potential limitations in terms of bioavailability, stability, as well as target efficacy. In the context of ocular disease, local administration of α-crystallin offers to circumvent challenges otherwise faced by systemic administration, and studies do emphasize this efficacy with considerable success. Intravitreal administration of α-crystallin in rats was found to result in a significant reduction of death of axotomized RGCs on intraorbital optic nerve crush [88]. Supplementation of α-crystallin was able to attenuate the expression of TNFα and nitric oxide synthase (iNOS) in microglia from rats following exposure to LPS and optic nerve crush, highlighting its anti-inflammatory properties [89]. Further investigation revealed the efficacy of repetitive intravitreal administration of α-crystallin to promote RGC survival as well as attenuate expression of pro-inflammatory cytokines otherwise secreted by Müller cells in response to optic nerve injury [90]. In another independent study, intravitreal administration of αB-crystallin was also shown to reduce ischemia-reperfusion injury mediated loss of retinal ganglion cells [91]. Intravenous administration of αB-crystallin was shown to promote recovery of mice retina following Anterior Ischemic Optic Neuropathy (AION) by suppressing neuroinflammation and promoting glial survival [92].

## 8. Conclusions

Altogether, our current understanding of the efficacy of both systemic and localized administration of α-crystallin offers considerable promise in its application to alleviate pathologies characterized by acute neurodegeneration. In contrast, data from studies investigating aging and chronic neurodegeneration assert the need for enhanced protein therapeutics as effective neuroprotective agents. Recombinant αA- and αB-crystallins harboring mutations eliciting a gain of function phenotype-enhancing the chaperone and anti-apoptotic function, can offer better neuroprotection. Studies from our lab have shown that ‘conditioned’ growth media from glial cells overexpressing αA-crystallin was shown to protect R28 neuronal cells from nutrient deprivation-induced apoptotic stress [22]. Media obtained from glial cells expressing the T148 phosphomimetic T148D was shown to have a greater effect in promoting cell survival in comparison to the wild-type protein and the non-phosphorylatable control, T148A. The study for the first time established the inherent role of PTM enhancing the chaperone function of αA-crystallin in the retina. Studies investigating the structure function relationship of αA-crystallin have also indirectly highlighted the cytoprotective efficacy of these ‘enhanced’ proteins. Biochemical analyses of a mutant of αA-crystallin, αA-R21Q, associated with congenital cataract revealed that this gain of function mutation enhanced the survival of ARPE-19 cells under oxidative stress [93]. R21Q was also shown to act as a ‘suppressor’ mutation, effectively alleviating the deleterious effects otherwise observed with αA-G98R crystallin [94,95,96,97]. Our most recent study highlighted that expression of T148D crystallin in glial cells prevents their activation under diabetic stress, notably by the suppression of pro-inflammatory cytokines [43], effectively participating in dampening diabetes-induced overt inflammation. Collectively, these observations support a viable therapeutic strategy involving the external administration of purified functionally enhanced αA-crystallins to promote neuronal cell viability to counteract chronic neurodegenerative stress.

## Figures and Tables

**Figure 1 antioxidants-10-01001-f001:**
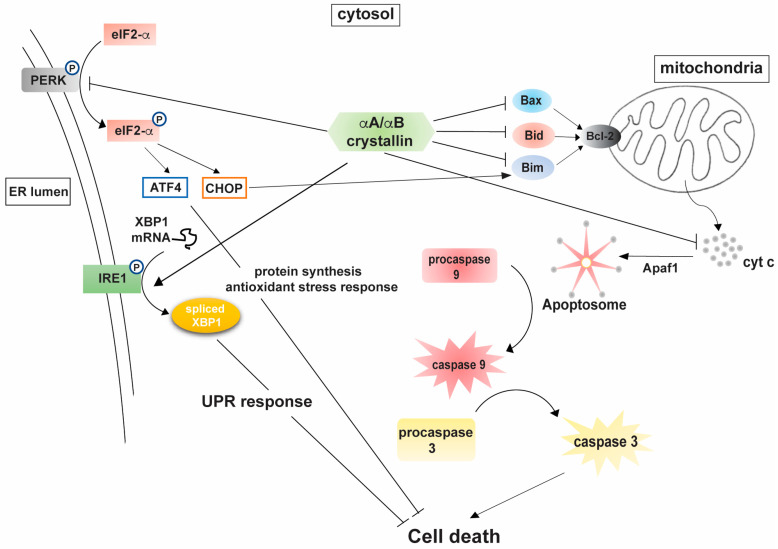
Mechanistic analyses of α-crystallin mediated cytoprotection. Abbreviation list: cyt c, cytochrome c; Apaf1, apoptotic protease activating factor 1; eIF-2α, eukaryotic translation initiation factor 2α; ATF4, activating transcription factor 4; CHOP, C/EBP homologous protein; XBP1, X-box binding protein; IRE1, Inositol requiring enzyme 1; PERK, Protein kinase R-like ER kinase; ER, endoplasmic reticulum; UPR, unfolded protein response.

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
