# Peer review of "Therapeutic Potential of α-Crystallins in Retinal Neurodegenerative Diseases"

_antioxidants, 2021, doi:10.3390/antiox10071001_

Round 1
Reviewer 1 Report
This article reviews the therapeutic potential of chaperone and α-crystallins in retinal neurodegenerative disease.
The major concern is many of their review studies focus on the α-crystallins in human lens epithelial cells and others. Why the title just mentions retinal neurodegenerative disease?
Minor:
1.Line 81 & 110 & 114…: The a-crystallins, comprise of aA and aB subunits. à α?
Author Response
Comments by reviewer 1:
This article reviews the therapeutic potential of chaperone and α-crystallins in retinal neurodegenerative disease. The major concern is many of their review studies focus on the α-crystallins in human lens epithelial cells and others. Why the title just mentions retinal neurodegenerative disease?
Thank you for your perspective on our recent work. Ever since their discovery of their chaperone properties, a multitude of studies on a-crystallins were focused on their function in maintaining the transparency and the refractile properties of the organ, and eventually transcended to investigation of their anti-apoptotic function in cell model systems. As summarized in our review, there is overwhelming evidence that supports the potential of the use of a-crystallins to exploit their anti-apoptotic function. This application is especially beneficial in promoting viability of post mitotic cells such as neurons to circumvent problems associated with aging. Retinal neurodegeneration is the leading cause of blindness in the working age population today. In the recent years, work from our lab and others has highly supported the involvement of aA- and aB-crystallin in retinal homeostasis. Both proteins have been shown to exhibit neuroprotective efficacies in cell and animal models of retinal disease. Furthermore, evidence also suggests that the proteins can be taken up by the cells on exogeneous supplementation, with minimal to no immunogenic effects. The aim of this review was therefore to summarize our current understanding of the anti-apoptotic function of a-crystallins from diverse in vitro studies and animal models and expand the possibility of using these valuable protein candidates against retinal neurodegeneration. This point has been further emphasized in the abstract and introduction of the revised manuscript.
Minor:
- Line 81 & 110 & 114…: The a-crystallins, comprise of aA and aB subunits. à α?
The suggested changes have been included in the revised manuscript.

Reviewer 2 Report
This study reviews on the clinical significance of α-crystallins for retinal neurodegenerative diseases. It is a well written manuscript which I found truly interesting. The authors cover this subject efficiently by presenting different aspects and perspectives. They also demonstrate research data accurately and explain the scientific ideas in an understandable yet precise way.
The only comment I have is regarding the bibliography, considering that only 10 out of the 91 references used on this study have been published at the last 5 years. I believe that the manuscript would be greatly improved if the authors could cite/include more recent articles as references. Therefore, I propose the publication of this manuscript after minor revisions.
Author Response
Comments by reviewer 2:
This study reviews on the clinical significance of α-crystallins for retinal neurodegenerative diseases. It is a well written manuscript which I found truly interesting. The authors cover this subject efficiently by presenting different aspects and perspectives. They also demonstrate research data accurately and explain the scientific ideas in an understandable yet precise way.
The only comment I have is regarding the bibliography, considering that only 10 out of the 91 references used on this study have been published at the last 5 years. I believe that the manuscript would be greatly improved if the authors could cite/include more recent articles as references. Therefore, I propose the publication of this manuscript after minor revisions.
Thank you for your valuable suggestion for bettering our work. Several key recent references have been added to the revised manuscript.
